# Improved Light Field Compression Efficiency through BM3D-Based Denoising Using Inter-View Correlation

**DOI:** 10.3390/s21092919

**Published:** 2021-04-21

**Authors:** You-Na Jin, Chae-Eun Rhee

**Affiliations:** 1Department of Electrical and Computer Engineering, Inha University, Incheon 22212, Korea; youna_0712@naver.com; 2Department of Information and Communication Engineering, Inha University, Incheon 22212, Korea

**Keywords:** multi-view, denoising, block matching 3D collaborative filtering (BM3D), video compression, high efficiency video coding (HEVC)

## Abstract

Multi-view or light field images have recently gained much attraction from academic and commercial fields to create breakthroughs that go beyond simple video-watching experiences. Immersive virtual reality is an important example. High image quality is essential in systems with a near-eye display device. The compression efficiency is also critical because a large amount of multi-view data needs to be stored and transferred. However, noise can be easily generated during image capturing, and these noisy images severely deteriorate both the quality of experience and the compression efficiency. Therefore, denoising is a prerequisite to produce multi-view-based image contents. In this paper, the structural characteristics of linear multi-view images are fully utilized to increase the denoising speed and performance as well as to improve the compression efficiency. Assuming the sequential processes of denoising and compression, multi-view geometry-based denoising is performed keeping the temporal correlation among views. Experimental results show the proposed scheme significantly improves the compression efficiency of denoised views up to 76.05%, maintaining good denoising quality compared to the popular conventional denoise algorithms.

## 1. Introduction

Multi-view images or light field data, acquired from a multi-camera array by capturing a plurality of images of different viewpoints, are used in various applications, including the synthesis of a high-resolution image with a high dynamic range [1] or the reconstruction of an occluded area through synthetic aperture photography (SAP) [2]. The light field information of dense multi-view images is frequently used in such applications as post-refocusing [3], super-resolution [4], occlusion reconstruction [5] and depth estimation. Recently, light field-based virtual reality (VR) recognizes an interactive and immersive “walk-around space” by rendering an image of an arbitrary viewpoint through a combination of pixels [6,7]. During the image capturing step, noise can be generated for various reasons. Noise appears in various ways due to various causes, such as the amount of light reaching the camera sensor or reflecting off the dust in the air. This type of noise can arise in far more diverse patterns in multi-view images than in single-view images. In particular, since the multi-view is photographed from multiple points or multiple cameras, the pattern of noise generated may be more diverse. It takes a very long time to remove the noise generated in a large amount of multi-view. The above has been added to the manuscript. When a user walks around in a light field or a multi-view based virtual space wearing a head-mounted display (HMD), the display is close to the eyes such that noise may severely deteriorate the quality of the experience. In addition, high-quality multi-view VR contents with a resolution exceeding 4 K require far more overhead for the storage and transmission of data compared to the conventional high-resolution images or the popular 360-degree video. Noise decreases the temporal correlation of the pseudo sequence, consisting of multi-view images (hereinafter referred to as multi-view sequence); therefore, the compression efficiency is reduced and transmission in an insufficient network bandwidth becomes difficult. As a result, denoising is a prerequisite to produce multi-view-based image contents for next generation immersive media to satisfy both the expected image quality and the compression efficiency.

A tremendous amount of works has been done in an effort to reduce noise in images. Table 1 summarizes the previous works. These research efforts have been extended from single-view image to videos and multi-view images. For denoising in single-view images, many studies based on the non-local mean (NLM) [8] algorithm have been suggested. Block-matching and 3-dimensional (3D) filtering (BM3D) [9,10], referring to an enhanced version of NLM, show excellent performance, but the processing time can be very long due to high computational complexity. Recently, denoising using a convolutional neural network (CNN) has been studied [11,12,13,14]. Studies of a self-guided network (SGN) [11] and FFDNet [14] focus on computational efficiency considering the trade-off between the inference speed and the denoising performance. Research efforts on removing real noise, as opposed to synthesized gaussian noise, are also being attempted [12]. Lefkimmiatis et al. construct two networks, a CNN layer as a core component and a layer for the non-local self-similarity properties of natural images, to handle a wide range of noise levels [13].

In studies involving videos, denoising techniques are based on the feature of consecutive frames or views having a high correlation. The Video BM3D (VBM3D) algorithm [15] increases the denoising quality by performing block matching not in a single image but in temporally neighboring frames. In the VBM4D algorithm [16], the motion vectors (MVs) between frames are tracked to obtain the trajectories by which to identify and group similar blocks. As filtering is conducted on the blocks in the trajectories, the temporal consistency can be fairly well maintained. However, because the tracking is limited to the horizontal and vertical directions, the denoising performance is poor for multi-view images with complicated textures. Execution times are much longer than in BM3D because the multi-view geometry is not actively considered. Research involving the grouping of similar patches using the optical flow shows denoising results that exceed that of the VBM4D, but the computational complexity is also very high [17]. Video noise is removed by training information between one frame and multiple frames [18], and a tensor approach was applied in a video denoising method called sparse and low-rank tensor (SALT) [19].

As multi-view applications have attracted attention, denoising research in this area is also increasing. Miyata et al. [20] propose an algorithm for the rapid denoising of multi-view images using the NLM with a plane sweeping (PS). In work by Zhou et al. [21], one of multiple views is set as the target view. The 3D focus image stack for the target view is made from other views. The disparity map obtained from the 3D focus image stacks is then used to generate a clean target view. When a densely arranged 2-dimensional (2D) camera array is used, a highly accurate depth map can be obtained from the 3D focus image stack. However, it does not work well for views with severe noise or wide view distances. Many multi-view algorithms using a CNN have also been proposed. Multi-view image denoising algorithm based on convolutional neural network (MVCNN) [22] that represents an enhancement of earlier work was also proposed [21]. Noisy 3D focus image stacks are fed into a denoising convolutional neural network (DnCNN) [23] and a disparity map is calculated from the denoised 3D focus image stacks. However, these types of approaches in which multiple views are used as references for one target view are aimed at denoising a single-view image utilizing the abundant information, but they do not take full advantage of the structural characteristics of multi-view images. A long time is required to remove the noise from all the views, because the process should be repeated as many times as the number of views. Meanwhile, light field denoising by sparse 5-dimensional (5D) transform domain collaborative filtering (LFBM5D) is a noise removal technique for light field (LF) images captured with a 2D camera array [24]. It performs patch search and grouping operation in both the angular and spatial domains targeting light field images that are densely acquired. The angular domain refers to sub-aperture images, whereas the spatial domain refers to a single image. The denoising performance is improved by securing many similar patches from many sub-aperture images. LFBM5D has also been applied to remove artifacts in super resolution studies of LF images [25]. Furthermore, Chen et al. used a CNN to remove noise from LF images based on an anisotropic parallax analysis (APA) [26]. For real-time LF denoising, 4-dimensional (4D) linear and shift-invariant hyper fan filters have been proposed and implemented in hardware [27]. However, the correlation among the denoised multi-view sequences is not maintained because compression was not considered. Consequently, the results showed significantly decreased compression efficiency.

**Table 1 sensors-21-02919-t001:** Previous works of denoising algorithms.

Target Applications	Denoising Algorithms
Single view	NLM [8]
	BM3D [9,10]
	SGN [11]
	Real image denoising [12]
	Universal denoising networks [13]
	FFDNet [14]
Video	VBM3D [15]
	VBM5D [16]
	With optical flow estimation [17]
	Frame-to-frame training [18]
	SALT [19]
Multi-view	NLM with PS [20]
	With occlusion handling [21]
	MVCNN [22]
	LFBM5D [24,25]
	APA [26]

An epipolar plane image (EPI) is a typical means of expressing the structure of multi-views. In an EPI, the relative motion between cameras or objects is expressed as lines that have different slopes depending on the depth. In comparison with regular photo images, an EPI has a very well-defined structure and there have been many studies of EPI estimation [28,29,30,31,32,33,34,35]. Depth estimation is the typical example in which an EPI is utilized actively. Zhang et al. proposed the EPI-based depth estimation that utilized a spinning parallelogram operator (SPO) for light field camera images with noise and occlusion [30]. Meanwhile Calderon et al. use a two-step EPI estimation for light field images [31]. The initial estimation, conducted using a pixel by pixel approach, is done first. The aggregation process is then performed in order to ensure the same disparity for the same object. In the field of video coding, EPI-based motion estimation (ME) has been widely adopted. Li et al. estimate the dense disparity at a high accuracy by using the structural information obtained from a light field [32]. Lu et al. increase the video coding efficiency by using the epipolar geometry in multi-view images captured from various angles for an object at the center, where the epipolar line of a multi-view image is obtained, after which the disparity vector is acquired by searching over the line [33]. Similarly, other studies [34,35] also propose EPI-based disparity or ME schemes. However, a study to combine image processing such as denoising with an EPI structure has not been seriously attempted thus far.

This paper takes advantage of the structural characteristics of linear multi-view images to increase the denoising speed and performance as well as to improve the compression efficiency of the denoised multi-view sequence. The key contributions of the proposed scheme are as follows. First, the 3D filtering and the multi-view geometry-based EPI estimation are effectively combined to speed up the multi-view denoising. Second, to increase the accuracy of the EPI estimation in the noisy multi-view, the consistence of disparity vector is considered both in the inter- and intra-views. To do this, the search is performed in units of a 1-dimensional (1D) window and the search range boundary is carefully determined in the inter-views. Third, compression-friendly EPI is estimated instead of pursuing the true EPI. This is to reflect the characteristics of the ME of video coding which performs a block-based search and considers the trade-off between bitrate and accuracy (which equals quality). The proposed scheme effectively increases the compression efficiency when the noisy multi-view images must be stored and transmitted. 

The remainder of this paper is organized as follows. Section 2 describes the entire flow of the proposed algorithm, which combines EPI and BM3D. Section 3 explains the details of the EPI estimation process and the experimental results are provided in Section 4. Section 5 gives the conclusions.

## 2. Overview

The flow in Figure 1 shows the BM3D-like denoising process. The left side gives the single-view-oriented techniques of BM3D. In the first search step, the non-local search is carried out to find the patches that are similar to the target patch T. The patches found are grouped. In the second filtering step, the individual groups undergo 3D transformation and hard-thresholding to remove high-frequency signals, assuming that noise is present at the high frequency. Here, the weight is determined according to how much the noise is removed. After the inverse transformation, the filtered patches are aggregated according to weight. This produces the denoised view images. 

There are some difficulties when BM3D is directly applied to multi-view images. First, the searching and grouping of similar blocks through block matching take too much time. This processing time increases in proportion to the number of views in multi-view images. Second, 2D block matching based on the sum of absolute difference (SAD) results in low accuracy in a noisy image. Third, view-by-view denoising cannot maintain consistency among the denoised views, resulting in a reduction of the multi-view sequence compression efficiency. VBM3D and VBM4D, based on BM3D, also have similar limitations.

The right side in Figure 1 shows the proposed scheme. In the multi-view images obtained from a linear 1D camera array, corresponding pixels or patches are found at the same height of all views, that is, in the epipolar line. Therefore, the general 2D blocking matching is re-placed with epipolar line searching (ELS) in the unit of a 1D window. In the case of a 1D window (search window), a relatively small range of 11 pixels is used in consideration of complexity and execution time. Given the target pixel, a set of the corresponding pixels from multiple views forms a straight line, which is hereinafter referred to as the corresponding pixel line (line_cor-pixel_). Grouping is performed in a 2D patch unit along the line_cor-pixel_ and the eight patches are transferred to the filtering step because the computational complexity of the 3D transformation increases sharply according to the number of patches. The subsequent steps are the same as for BM3D. In the last step, the inter-view aggregation is performed on the multiple views. 

The key ideas with regard to how the proposed combination of patch searching and 3D transform resolves the difficulties explained in existing research are as follows. 

Searching time accounts for nearly half of the total time in BM3D-like approaches such as BM3D, VBM3D, VBM4D and LFBM5D. In this paper, both the search area and the difference calculation are changed from 2D to 1D considering the characteristics of 1D linear geometry. Thus, the searching speed is increased.The naïve block-matching search used in the conventional BM3D-like approaches does not work well with noise. For an efficient search in the noisy views, the proposed 1D window-based search using EPI, which accurately reflects the characteristics of multi-view, reduces the risk of incorrect patches. In addition, the maximum disparity is reflected in the ELS range of the initial views (denoted as SR_1step_ hereinafter) as meaningful searching guidance for noisy images. Subsequently, the remaining views limit the search range (denoted by SR_2step_ hereinafter) around the best point found in the previous views to prevent an incorrect outcome due to noise.Most existing denoising algorithms simply consider the SAD in block search process. However, the proposed ELS plays the role of pre-processing for the block-based ME to be carried out for the denoised multi-view compression. When searching for the correspondence in each view, the slope similarity of the line_cor-pixel_ with adjacent pixels as well as the pixel difference is considered together to reflect the rate-distortion optimization of ME. This has a positive effect in the subsequent compression stage of denoised multi-views.Aggregation using weighted summation helps denoised results to be spatially consistent within the image in BM3D-based approaches. Hereinafter, this is referred to as spatial aggregation. In this paper, both spatial and temporal aggregations are adopted. The correlation between the views is increased by performing inter-view level temporal aggregation from the filtering results for the patches of each view. This contributes to increasing the compression efficiency.

## 3. Proposed Compression-Friendly Denoising 

### 3.1. Fast and Noise-Resistanti EPI Estimation

ELS as proposed in this paper consists of three steps for initial views, remaining views and unmatched pixels. 

In the first step, EPI estimation is conducted with the first and second views from the left side of the given multi-views. In Figure 2, *v*_1_ and *v*_2_ represent the first and second views, respectively, whereas X and Y correspondingly denote the width and height of the views. *T*_1_, a pixel of *v*_1_, is set as the denoising target pixel. To perform ELS on *v*_2_, searching begins with the reference pixel *r*_2_ located at coordinates identical to those of *T*_1_. In a noisy view, EPI estimation in pixel units with respect to the Y-component may be inaccurate. To take the adjacent pixels into consideration together, ELS is performed in a 1D window unit with a window size of (2w + 1). Considering *v_2_* is located to the right of *v*_1_, ELS is performed toward the left of the reference pixel *r_2_* in the range of the *SR_1step_* which is set to the maximum disparity that can be varied depending on the distance between the camera and the closest object in multi-views. The pixel *c*_2_ with the minimum cost, found in *v*_2_, becomes the correspondence of *T*_1_. The horizontal difference between *T*_1_ and *c*_2_ is referred to as the disparity vector (DV), and the line connecting the corresponding pixels, shown in Figure 2, is referred to as the line_cor-pixel_. Repeating the same process with respect to the X × Y area of *v*_1_ gives EPI of the two views.

There are two criteria for judging the correspondence in ELS. First, the difference from the target pixel should be small so as to increase the denoising effect by securing similar patches in the filtering step of Figure 1. Second, the DV should be as similar as the MV which will be obtained from ME in the video compression to be conducted after denoising. Equation (1) briefly shows J_motion_, which is the cost function when the ME searches for the optimally matched block in video compression such as H.264 [36] or HEVC [37]. D_(T, R)_ represents the difference between the current target block, T, and the reference block, R, whereas R_MVD_ indicates the number of bits occurring when coding an MV. This is usually replaced by the MV difference between the current and predicted MV. λ_motion_ represents the Lagrangian multiplier [38,39]. The minimum cost is determined considering both the rate and distortion.
J_motion_ = D_(T, R)_ + λ_motion_·R_MVD._(1)

For ME friendly ELS, the cost function in (2), which is designed by modifying (1), is used to search for the correspondence. D_(T, R)_ is the sum of the squared errors between the current window containing a target pixel and the reference window on the epipolar line of the next view. R_DVD_ is the difference between the current and predicted DVs. The predicted DV is obtained from the median DV of the top and left pixels of the current view already searched, whereas the current DV is the disparity vector between the current and reference windows. In many existing block matching-based noise denoising algorithms, only the difference between blocks is used as a metric to determine the block similarity. This is because filtering on similar blocks greatly influences the denoising performance. However, this paper noted that in ME of video coding, consistent MV along with the similarity of pixel values is an important criterion for block matching. Therefore, R_DVD_ is added in (2) to reflect such spatial correlation of video coding to block matching of denoising. By considering the pixel difference and R_DVD_ together, it is possible to achieve not only denoising performance but also compression efficiency in the multi-view compression due to enhanced spatial correlation. Also, a point with a small R_DVD_ value is preferred to follow the condition in ME which considers the rate-distortion trade-off as shown in (1).
cost = D_(T, R)_ + λ·R_DVD_.(2)

In the second step of ELS, the EPI of all remaining views, starting with *v*_3_, is obtained. In Figure 3, the specific epipolar lines of the four views (*v*_1_, *v*_2_, *v*_3_ and *v*_4_) are stacked. Each square denotes a pixel. The gray and white pixels represent the background and the foreground in the view, respectively. The box with the thick line represents the 1D window with w = 3 to be used in searching. In the first step shown in Figure 2, the correspondence *c_2_* has already been found in *v_2_* for the target pixel *T*_1_ of *v*_1_. If *d*_1*_*2_, the DV of *c*_2_, is used as the predicted DV, the initial reference pixel *r*_3_ for ELS in *v*_3_ is obtained by subtracting *d*_1*_*2_ from the x coordinate of *c*_2_, which is *c*_2_(x). The cost is calculated for the range of *SR*_2*step*_ pixels indicated by the dotted lines containing *r*_3_. Out of these, the pixel having the lowest cost is determined using (2) to be the correspondence *c*_3_. For ELS of *v*_4_, the DV *d*_2*_*3_ is used as the predicted DV. The initial reference pixel *r*_4_ is calculated from *c_3_*(x) − *d*_2*_*3_. By using the DV in the directly prior view as the predicted DV, setting *SR*_2*step*_ such that it is very small is reasonable. Thus, the search range of the second step can be significantly reduced compared to the range of *SR*_1*step*_ in the first step. The same process is carried out for all views except *v*_1_ and *v*_2_.

After the first and second steps, the drawing line_cor-pixel_ on all views given the target pixels of *v*_1_ is complete. However, in other views except *v*_1_, there may be non-corresponding (*nc*) pixels which do not belong to any group. If these *nc* pixels are excluded from filtering, the denoising performance will decrease. To solve this, *nc* pixels are set as new target pixels. Then, ELS is performed for them. Since this process is done to include *nc* pixels in 8 × 8 patch-based filtering, the target pixels are determined by sampling rather than using all *nc* pixels. ELS is performed for the views *v*_*t*−1_ and view *v_t+_*_1_, which are respectively the previous view and the next view of the *v_t_* to which the target pixel belongs. Here, the predicted DV is obtained by referring to the line_cor-pixel_ that is closest in the horizontally left direction of the target pixel in *v_t_*. Based on the predicted DV, initial reference pixels are determined in *v_t−_*_1_ and *v_t+_*_1_, and searching is performed in the adjacent range of *SR*_2*step*_. The ELS processes are conducted toward the right and left from the reference pixels in *v_t−_*_1_ and in *v_t+_*_1_, respectively.

### 3.2. Temporal Aggregation

In this paper, temporal aggregation is additionally applied along with spatial aggregation to improve the correlation between multi-views further. Spatial aggregation is the same as the existing BM3D-like approach. In a single view, denoised patches are returned to their position through weighted summation reflecting the degree of noise removal. In temporal aggregation, a similar operation is performed at the inter-view level. The application of temporal aggregation is determined by means of a binary confidence judgment. This binary confidence is based on the cost in (2) used in the EPI estimation process. A low cost indicates that the difference and disparity variance of the patches on the line_cor-pixel_ are sufficiently small, meaning that the confidence of the estimated EPI is high. Therefore, when the cost value is lower than the threshold, temporal aggregation is performed. The threshold is set empirically. Temporal aggregation is performed between views by reflecting weights according to denoising results of patches from each view, contributing to the improvement of the compression efficiency. 

## 4. Performance Evaluation

The proposed scheme in this paper is evaluated in terms of denoise performance, compression efficiency and computational complexity. Figure 4 shows ten datasets used in the experiment. *Champagne*, *Pantomime* and *Dog* have 80 views, whereas *Wall* and *Piano* have 96 and 64 views, respectively. *Kendo* and *Balloons* consist of four views [40,41]. *Ohta*, *Sideboard* and *Cotton* datasets are additionally used to test the performance when the proposed scheme is applied to not only 1D but also 2D multi-view. *Ohta*, *Sideboard* and *Cotton* have 25 and 81 views, respectively [42]. Noise is added to each test sequence using the additive white gaussian noise (AWGN) of sigma values of 15, 25 and 35. For HEVC encoding, pseudo sequences are created from multi-view images and HM-16.10 reference software [43] with the low-delay-P-Main (LD) configuration. Quantization parameters (QPs) are set to 22, 27, 32 and 37. All the experiments are conducted on an Intel i7-8700K running 3.70 GHz and 64 GB memory. The superiority of the proposed scheme is compared with BM3D [9], VBM4D [16], MS-EPLL [44], 3D-focus [21], MVCNN [22] and LFBM5D [25]. The codes used for the proposed scheme and for performance comparison are uploaded to https://github.com/Digital-System-Design-Lab/Compression-Aware-Multi-view-Denoising.git (accessed on 21 April 2021) [45]. In the configuration of BM3D, VBM4D and LFBM5D, only hard thresholding is applied without Wiener filtering steps. The patch size is 8 × 8 pixels. The maximum number of patches are set to 8 for *Champagne*, *Pantomime*, *Dog*, *Wall* and *Piano*, and 4 for *Balloons*, *Kendo*, *Ohta*, *Sideboard* and *Cotton*. The search area is 32 pixels horizontally and vertically with a one-pixel spacing distance. Spacing is a major factor that determines the trade-off between processing time and denoising performance. Commonly, spacing = 3 is used to reduce complexity, whereas spacing = 1 is used when denoising performance is important. The settings used in this experiment follow the previous BM3D-like approaches.

Table 2 shows the denoise performance when using the conventional denoising algorithms and the proposed scheme. Only *Champagne*, *Pantomime*, *Dog*, *Wall*, *Piano*, *Balloons* and *Kendo*, which are horizontally linear multi-view datasets acquired with a 1D camera array, were tested. From the third to ninth columns, the quality of the denoised multi-view images was compared to the original clean images using the peak-signal-to-noise ratio (PSNR). Given a noise-free m × n monochrome image I and its noisy approximation K, MSE is defined as follows:
(3)$$\mathrm{MSE}=\frac{1}{m\;n}\sum_{i=0}^{m-1}\sum_{j=0}^{n-1}{\left[I\left(i,j\right)-K\left(i,j\right)\right]}^{2}$$

The PSNR can be calculated as follows:(4)$$\mathrm{PSNR}=10\cdot \log_{10}\left(\;\frac{{\mathrm{MAX}}_{I}^{2}}{\mathrm{MSE}}\;\right).$$

Among seven algorithms, MS-EPLL and the proposed scheme show the highest PSNR when sigma = 15, whereas the MS-EPLL is best when sigma = 25 and 35. VBM4D is about 0.85 dB lower than BM3D even though more views are used for denoising. In block matching of VBM4D, simplified searching decreases the denoising performance because there is a risk of finding wrong patches in a complex image. MS-EPLL shows higher PSNR than BM3D and proposed scheme in most test sequences. However, blur due to filtering is noticeable. Therefore, its PSNR is lower than the proposed scheme for the small sigma or for the high detail views such as *Piano* and *Dogs*. Using 3D-focus is basically an algorithm that assumes a light field camera to shoot densely. Thus, its performance is good in multi-view, which is very tightly shot such as *Piano*. However, in the general multi-view with a wide camera distance, disparity maps are not well estimated, resulting in poor noise reduction. MVCNN also suffers from similar problems with 3D-focus. As an incorrect disparity map is generated, a strong blur occurs. Since LFBM5D does not fully utilize horizontally linear structural information, the accuracy is somewhat low when searching for similar patches in noisy views. This problem is more pronounced in datasets such as *Balloons* and *Champagne* with wider view distance. As the sigma of AWGN is larger, blur appears at the edge and serious artifacts such as color contamination occur due to grouping of wrong patches. As shown in Table 2, it was confirmed that the PSNR is lower than that of the proposed scheme which fully utilizes a multi-view structure. From the tenth to fifteenth columns of Table 2, the compression efficiency of the denoised multi-view images is measured with the Bjontegaard Delta Bitrate (BDBR) [46], which is calculated using the bitrate used during the transmission and the PSNR between already denoised input views in the encoder side and the reconstructed views in the decoder side. The BDBR in Table 2 show the reduced bitrate of various denoising schemes compared to BM3D. In Pantomime, having a wide range of black background and little object movement, both VBM3D and the proposed scheme which consider the correlation between views, achieve a bitrate reduction compared to BM3D. However, in other sequences, the proposed scheme shows the highest compression efficiency. For complex and noisy sequences, the BDBR performance of the proposed scheme is much improved. In VBM4D, it is difficult to denoise the multi-view with high details or wide camera distance. Thus, the compression efficiency is also poor. In MS-EPLL, as the sigma value increases, the compression efficiency increases significantly. Although temporal correlation among views is not considered in MS-EPLL, the increased similarity between views from the severe blur can help video coding. In the case of 3D-focus, the compression efficiency is not good because the noise is not removed well and the color is distorted. In the case of MVCNN, the denoising performance is greatly improved compared to 3D-focus. The compression performance is also high due to the blurred result. The LFBM5D’s BDBR performance seems to be good, but it is due to blur caused by wrong patch grouping. This appears stronger as the sigma value increases. 

Table 3 shows the multi-view denoising performance as acquired from a 2D camera array in terms of the objective image quality and compression efficiency. This experiment shows that the proposed scheme can easily be extended to a 2D light field by estimating EPI horizontally as well as vertically. The PSNR value of the objective image quality measures the difference between the clean original multi-view and the denoised multi-view with each algorithm. In the case of BDBR, a multi-view denoised image with BM3D was used as a reference. Among a number of denoising algorithms for 2D multi-view, the performance of the proposed scheme was verified through a comparison with LFBM5D, which shows excellent denoising results. The 2D multi-view was converted to a spiral pseudo sequence and input to the HEVC encoder. The *Ohta*, *Sideboard* and *Cotton* datasets were used. *Ohta* has 5 × 5 = 25 views, whereas, in Ohta2, the views were down sampled to 3 × 3 = 9 views so as to assume an environment with a wide view distance. *Sideboard* and *Cotton* used 9 × 9 = 81 views. AWGNs with sigma = 15, 25 and 35 were used. The average PSNR of LFBM5D is 29.46 dB, whereas that of the proposed is 30.97 dB, which is approximately 1.51 dB higher compared to LFBM5D. In the case of objective image quality, it was verified that the proposed scheme outperforms LFBM5D in all datasets and under all sigma conditions. In the case of BDBR, the average BDBR of LFBM5D was −18.41%, whereas that of the proposed is −33.76%. LFBM5D shows high compression efficiency in datasets with narrow view distances, such as *Ohta, Sideboard* and *Cotton* in this case, but not in datasets with wide view distances, such as *Ohta2*. When the proposed scheme was applied, the BDBR performance was excellent for all datasets regardless of the distance between the views.

For a fair comparison with a deep learning based approach optimized for low resolution images, Table 3 and Table 4 show the PSNR and BDBR of denoised multi-views in four algorithms of BM3D, MS-EPLL, MVCNN and the proposed scheme when the resolution of seven datasets is reduced to 1/4. In the patch based denoising algorithm, the lower resolution increases the spatial complexity of the image patches, which can degrade the denoising performance. In Table 4, the average PSNR of each algorithm is 32.92 dB for BM3D, 33.36 dB for MS-EPLL, 22.25 dB for MVCNN and 33.18 dB for the proposed scheme. In Table 5, compared to BM3D, the average BDBR for each algorithm is +6.83% for MS-EPLL, −79.54% for MVCNN, and −25.78% for the proposed scheme. MVCNN has good compression efficiency due to excessive blur, but its PSNR value is very low. MS-EPLL even increases the BDBR value at low sigma value. As shown in Table 3 and Table 4, the proposed scheme is superior to other algorithms when considering both denoising performance and compression efficiency even in low resolution images.

Figure 5 and Figure 6 show the subjective quality of denoised multi-view densely acquired with a 1D camera array. After adding noise with sigma = 25 to *Kendo* and *Piano*, various denoising algorithms were applied. In the result of BM3D and the proposed scheme, the edge is clear, and the noise is removed quite well. In VBM4D, many noises are not removed properly. The probability of grouping the wrong patches is high because patch searching is done in the vertical direction. In the MS-EPLL, blurs are observed strongly in the human face in Figure 5 and the brick wall in Figure 6. Many details disappear. Inaccurate disparity estimation of 3D-focus not only degrades denoising performance but also causes color distortion. In the multi-view with a large camera distance such as *Kendo* in Figure 5, the disparity map of MVCNN is inaccurate and thus, blur is observed. In the case of *Piano* in Figure 6 which consists of dense views, the quality looks relatively clean, but it also has a lot of blurred edges. 

Figure 7 shows the subjective quality of *Ohta*’s denoised multi-view. Here, Figure 7a shows a clean view without added noise, whereas Figure 7b is the result of denoising with LFBM5D when the AWGNs with sigma = 15, 25, and 35 are added. Additionally, Figure 7c results when the proposed scheme is applied. Since LFBM5D did not find similar patches well, noise that could not be removed and broken edges were observed. These artifacts are more severe as the sigma value becomes higher and the distance between the view’s increases. Meanwhile, areas with small disparity values, such as background areas, are well denoised, but strong blurring is noticeable. Consequently, the PSNR performance is poor, but it is advantageous to increase the compression efficiency. In the case of what was proposed, because similar patches were well searched, the denoising performance was quite good with clear edges. 

Table 6 shows improved BDBR results when both spatial and temporal aggregation are used. The third column shows the results when only spatial aggregation is applied, whereas the fourth column presents the results when both spatial and temporal aggregations are applied together, which is identical to the fifteenth column in Table 2. When temporal aggregation is added, it can be seen that the compression efficiency is further improved by −10.54% in BDBR. 

Table 7 shows the execution time of the tested codes of each algorithm. For BM3D, two code types, each implemented in C++ [9] and Matlab [10], were used. The implementation of the two codes differs slightly, but the denoising performance of the C++ version is much better than that of the Matlab version. VBM4D [22], MS-EPLL [44], 3D-focus and MVCNN were implemented in Matlab, and LFBM5D and the proposed scheme were implemented in C++. In this experiment, the average execution time of the Matlab version of BM3D was set to 1.00 when processing for images with resolutions of 1024 × 768 and 1280 × 960, and the execution times of different algorithms were compared relatively. VBM4D, MS-EPLL, 3D-focus and MVCNN require 3.59 times, 1618.90 times, 127.33 times and 29.33 times longer execution compared to the execution times of the Matlab version of BM3D. Meanwhile, BM3D’s C++ version is 9.00 times longer, LFBM5D is 35.27 times longer, and the proposed scheme is 1.80 times longer. The proposed scheme achieves high denoising performance and compression efficiency while having a much faster execution speed than most other algorithms.

Table 8 shows the results of a comparison of the computational complexity required for denoising. First, for the block searching, which is the most time-consuming step during the denoising process, the second column in Table 8 is obtained from the denoising of the *Champagne* sequence and indicates the number of subtraction operations per view when 80 views 1280 × 960 in size are processed. For the block-matching in BM3D, the number of subtractions is calculated as *h* × *w* × *s_p_*^2^ × *s*^2^, where *h* and *w* denote the height and width, respectively. *s_p_*^2^ represents the patch size, whereas *s*^2^ represents the search area. For the block-matching in VBM4D, the number of subtractions comes to *h* × *w* × *h_t_* × *s_p_*^2^ × *s*. Here, *h_t_* is the number of patches found in other views considering the temporal correlation in VBM4D. 3D-focus selects the best five windows and additionally searches for similar windows around them. The required computations are *h* × *w* × *(5* × *s_p_*^2^
*+ s*^2^*)*. For the block-matching in LFBM5D, the number of subtractions is calculated as *h* × *w* × *s_p_*^2^ × *n_sim_*. *n_sim_* is a 2D search window to find similar patches within a single view. In the proposed EPI-based search, the total is *h* × *w* × *(s_w_* × *s*_1_ + *s_w_* × *s*_2_*)*. Here, *s_w_* denotes the 1D window size, whereas *s*_1_ and *s*_2_ represent the search range in the first and second steps, respectively. In summary, VBM4D and the proposed scheme show the highest and the lowest level of search complexity. Based on the 1K resolution image, computations for searching are calculated. CNN-based denoising is excluded. The amounts of computation for searching when using BM3D, VBM4D, 3D-focus and LFBM5D are approximately 1677, 15,859, 34 and 131 times larger, respectively, than that in the proposed scheme. In the third column, the filtering complexity is compared using the number of patches because the filtering time is proportional to the number of patches. The number of patches required per view is lowest with VBM4D and highest with 3D-focus. 

## 5. Conclusions

Until recently, many works to remove noise in images have been actively conducted, but the structural characteristics of multi-views cannot be effectively utilized. The proposed scheme in this paper improves the consistency of denoised multi-view as well as the performance of noise reduction by combining an excellent denoising algorithm BM3D with EPI reflecting the correlation of multi-views acquired from camera array. Experimental results also show that improved consistency has a significant positive effect on compression efficiency. In the future, additional intensive study is required through algorithmic and architectural co-development for fast processing of massive multi-view.

## Figures and Tables

**Figure 1 sensors-21-02919-f001:**
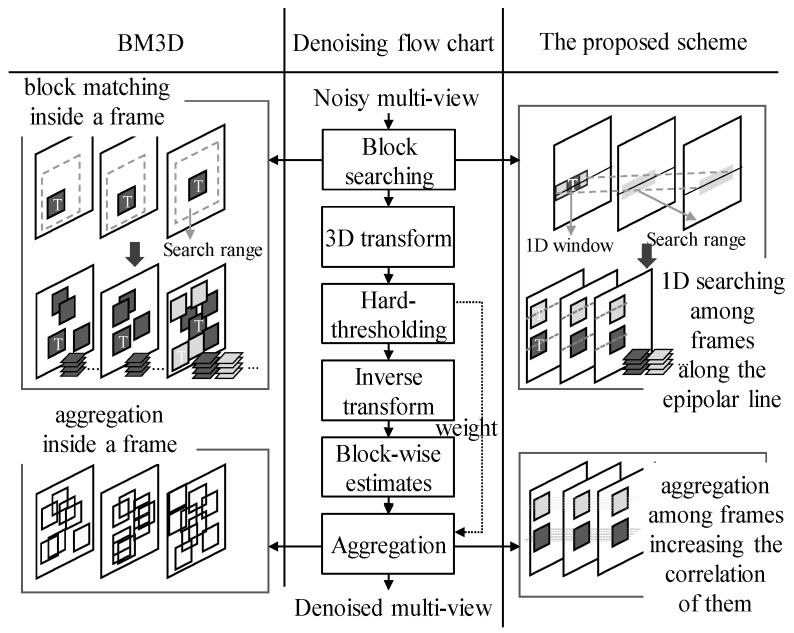
Flow chart of the proposed scheme.

**Figure 2 sensors-21-02919-f002:**
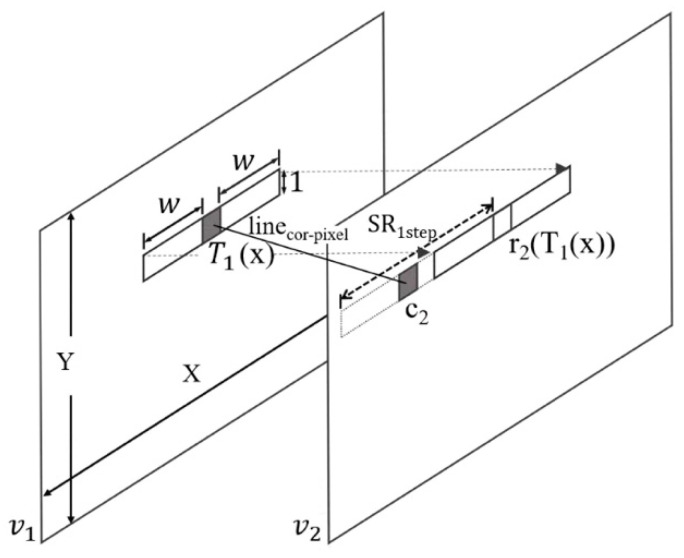
Initial EPI estimation to obtain the predicted slope of line_cor-pixel_.

**Figure 3 sensors-21-02919-f003:**
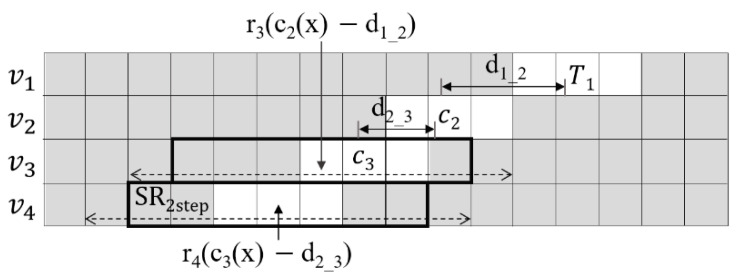
Fast search based on predicted slope.

**Figure 4 sensors-21-02919-f004:**
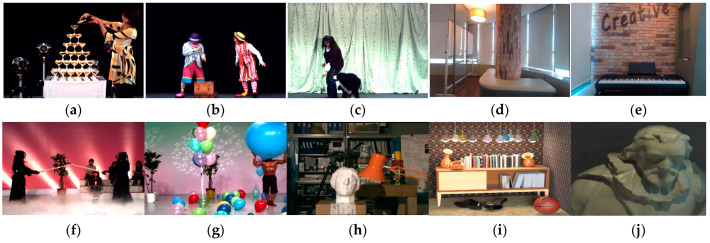
Dataset. (**a**) Champagne; (**b**) Pantomime; (**c**) Dog; (**d**) Wall; (**e**) Piano; (**f**) Kendo; (**g**) Balloons; (**h**) Ohta; (**i**) Sideboard; (**j**) Cotton.

**Figure 5 sensors-21-02919-f005:**
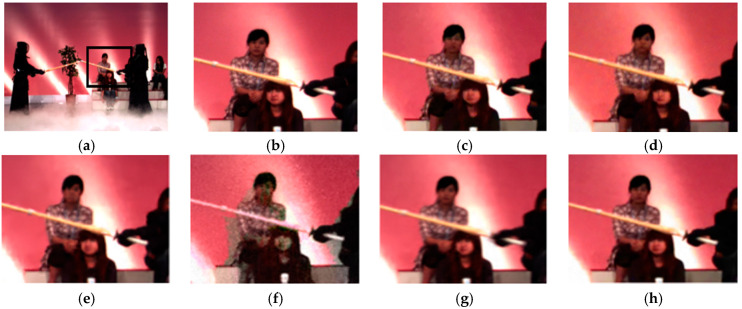
Subjective quality of *kendo*. (**a**) Full image of original; (**b**) original; (**c**) BM3D; (**d**) VBM4D; (**e**) MSEPLL; (**f**) 3D focus; (**g**) MVCNN; (**h**) proposed.

**Figure 6 sensors-21-02919-f006:**
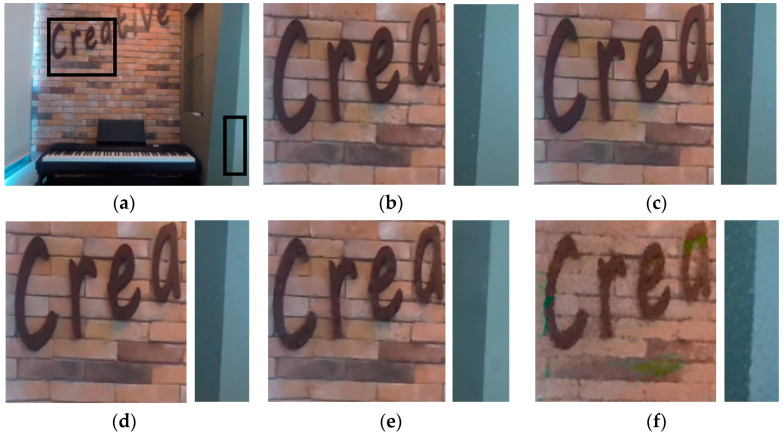
Subjective quality of *Piano*. (**a**) Full image of original; (**b**) original; (**c**) BM3D; (**d**) VBM4D; (**e**) MSEPLL; (**f**) 3D focus; (**g**) MVCNN; (**h**) LFBM5D; (**i**) proposed.

**Figure 7 sensors-21-02919-f007:**
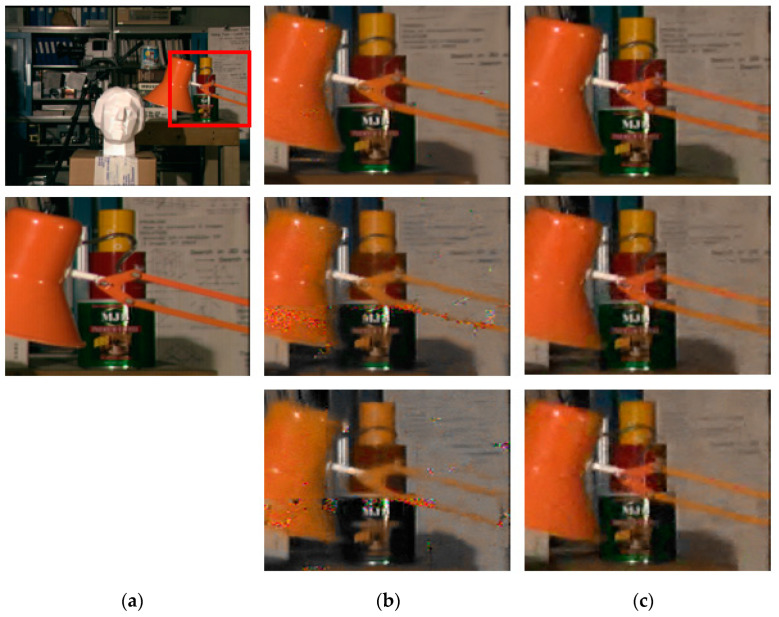
Subjective quality of Ohta. (**a**) Original image; (**b**) LFBM5D when sigma = 15, 25 and 35; (**c**) proposed when sigma = 15, 25 and 35.

**Table 2 sensors-21-02919-t002:** Comparison of performance for noisy multi-views.

Sequence	σ	PSNR (dB)							BDBR (%)					
		BM3D	VBM4D	MS-EPLL	3D-Focus	MVCNN	LFBM5D	Proposed	VBM4D	MS-EPLL	3D-Focus	MVCNN	LFBM5D	Proposed
Champagne (1280 × 960)	15	38.07	37.40	38.06	25.91	33.36	34.43	36.08	−9.22	+6.00	−22.47	−43.07	+9.00	−14.97
	25	35.23	34.72	35.78	25.06	31.08	30.90	33.93	−6.58	−6.73	−2.75	−51.97	+17.81	−20.98
	35	33.33	32.57	34.20	23.79	28.12	29.32	32.18	−0.88	−26.92	+12.83	−55.81	+9.94	−25.32
Pantomime (1280 × 960)	15	38.95	38.93	38.97	34.03	32.90	36.66	37.88	−51.43	+15.25	+36.92	−92.33	−49.85	−48.01
	25	36.20	36.46	36.81	29.45	31.72	32.43	35.39	−44.80	−7.90	+26.37	−62.47	−64.46	−58.29
	35	34.46	34.32	35.27	22.74	30.45	31.02	33.23	−54.02	−48.38	+62.43	−68.22	−77.88	−51.64
Dog (1280 × 960)	15	36.05	35.25	35.66	33.34	30.61	33.64	35.39	−19.83	+29.80	−33.55	−61.76	−37.74	−52.88
	25	33.18	32.84	33.45	30.05	30.28	30.50	33.11	−1.63	+4.37	+16.44	−64.24	−57.75	−60.86
	35	31.13	30.91	31.92	26.09	29.93	28.32	31.35	+47.82	−25.89	+13.24	−69.20	−61.28	−60.77
Wall (1024 × 768)	15	36.90	35.99	37.84	33.74	32.38	37.71	38.15	−8.08	+1.03	+189.44	−27.04	−55.88	−48.61
	25	33.81	33.42	35.47	31.60	32.07	34.60	35.38	+19.00	−36.03	+406.64	−33.56	−75.27	−65.42
	35	31.61	31.31	33.83	29.84	31.76	32.44	33.06	+148.10	−62.21	+167.57	−41.74	−83.41	−76.05
Piano (1024 × 768)	15	37.42	35.89	37.29	35.60	31.17	37.88	38.20	−13.02	+35.02	+522.99	−11.58	−50.05	−39.49
	25	34.46	33.14	34.90	32.38	30.87	34.27	35.35	+10.33	−4.02	+131.63	−19.55	−72.39	−55.59
	35	32.36	31.01	33.15	30.24	30.56	31.86	33.07	+130.98	−41.80	+127.47	−34.40	−79.18	−65.27
Kendo (1024 × 768)	15	37.73	35.88	38.14	33.47	31.23	36.28	36.71	+5.70	−23.47	+1891.84	−37.20	−39.03	−28.46
	25	34.82	33.13	36.21	30.14	30.95	33.11	34.68	+47.63	−43.35	+2218.42	−46.27	−41.51	−39.68
	35	32.69	30.97	34.81	27.83	30.66	30.71	33.05	+162.93	−58.75	+1902.10	−57.39	−41.56	−47.12
Balloons (1024 × 768)	15	36.84	34.56	36.91	32.12	31.13	34.86	36.88	+3.10	−14.68	+1071.66	−37.88	−21.85	−30.60
	25	33.96	31.92	34.77	28.97	30.54	31.73	34.03	+37.70	−30.12	+1220.68	−45.56	−30.30	−42.88
	35	31.92	29.86	33.27	26.80	30.14	28.42	32.30	+100.48	−47.50	+1060.70	−54.66	−35.73	−49.21

**Table 3 sensors-21-02919-t003:** Comparison of performance for noisy 2D multi-views.

2D Sequence	σ	PSNR (dB)		BDBR (%)	
		LFBM5D	Proposed	LFBM5D	Proposed
Ohta (384 × 288)	15	33.58	34.17	−40.00	−25.93
	25	29.79	31.21	−46.15	−30.24
	35	27.52	29.31	−43.21	−31.39
Ohta2 (384 × 288)	15	29.71	33.64	+40.45	−10.75
	25	26.45	30.99	+84.44	−11.18
	35	24.40	29.21	+129.02	−10.14
Sideboard (512 × 512)	15	34.08	33.92	−48.22	−20.90
	25	30.91	30.63	−60.90	−36.17
	35	28.77	27.15	−69.21	−40.67

**Table 4 sensors-21-02919-t004:** Comparison of denoising performance for noisy multi-views (low resolution).

Sequence	σ	PSNR (dB)			
		BM3D	MS-EPLL	MVCNN	Proposed
Champagne (640 × 480)	15	36.33	36.02	19.59	35.83
	25	33.70	33.37	18.97	33.36
	35	31.52	31.62	18.19	31.37
Pantomime (640 × 480)	15	37.35	37.22	24.00	37.20
	25	34.73	34.63	22.78	34.51
	35	32.40	32.88	21.25	32.27
Dog (640 × 480)	15	34.24	33.41	23.48	34.44
	25	31.81	30.84	22.15	31.85
	35	29.99	29.10	20.84	29.89
Wall (512 × 384)	15	37.34	36.30	24.57	37.76
	25	34.58	33.71	23.04	34.47
	35	32.57	32.04	21.37	32.11
Piano (512 × 384)	15	36.75	35.43	24.36	37.98
	25	33.88	32.70	22.77	34.66
	35	31.85	30.95	21.09	32.34
Kendo (512 × 384)	15	34.29	36.40	25.36	33.74
	25	30.93	34.01	23.66	30.92
	35	29.59	32.37	21.92	29.63
Balloons (512 × 384)	15	32.94	34.79	24.16	32.72
	25	31.20	32.22	22.61	31.27
	35	28.44	30.51	21.04	29.70
Cotton (512 × 512)	15	30.30	30.77	−47.96	−47.01
	25	29.48	30.56	−58.82	−63.30
	35	28.57	30.02	−60.34	−77.45

**Table 5 sensors-21-02919-t005:** Comparison of compression efficiency for noisy multi-views (low resolution).

Sequence	σ	BDBR (%)		
		MS-EPLL	MVCNN	Proposed
Champagne (640 × 480)	15	+15.94	−79.30	−8.97
	25	+7.31	−84.80	−15.52
	35	−13.03	−88.78	−20.52
Pantomime (640 × 480)	15	+36.13	−69.17	−21.72
	25	+19.17	−81.63	−28.50
	35	−16.65	−89.32	−33.21
Dog (640 × 480)	15	+80.70	−85.09	−28.40
	25	+50.12	−91.92	−35.59
	35	+0.67	−94.77	−35.37
Wall (512 × 384)	15	+43.53	−63.18	−21.32
	25	+7.02	−80.57	−30.45
	35	−30.01	−89.55	−29.00
Piano (512 × 384)	15	+67.55	−65.07	−36.21
	25	+18.96	−83.73	−48.27
	35	−29.12	−91.97	−47.86
Kendo (512 × 384)	15	+3.15	−58.87	−22.19
	25	−40.66	−80.85	−21.46
	35	−37.12	−70.09	−10.17
Balloons (512 × 384)	15	+14.29	−60.74	−8.91
	25	−7.93	−74.25	−16.88
	35	−46.64	−86.79	−20.83

**Table 6 sensors-21-02919-t006:** Comparison of performance for noisy multi-views when using both spatial and temporal aggregation.

Sequence	σ	BDBR (%)	
		Only Spatial	Spatial + Temporal
Champagne (1280 × 960)	15	−9.23	−14.97
	25	−19.32	−20.98
	35	−25.38	−25.32
Pantomime (1280 × 960)	15	−30.38	−48.01
	25	−46.67	−58.29
	35	−55.87	−51.64
Dog (1280 × 960)	15	−35.00	−52.88
	25	−16.43	−60.86
	35	−49.92	−60.77
Wall (1024 × 768)	15	−25.42	−48.61
	25	−40.50	−65.42
	35	−46.90	−76.05
Piano (1024 × 768)	15	−37.10	−39.49
	25	−57.28	−55.59
	35	−66.83	−65.27
Kendo (1024 × 768)	15	−22.93	−28.46
	25	−34.98	−39.68
	35	−39.48	−47.12
Balloons (1024 × 768)	15	−20.55	−30.60
	25	−31.83	−42.88
	35	−38.02	−49.21

**Table 7 sensors-21-02919-t007:** Comparison of processing time.

	Matlab	C++
BM3D	1.00	9.00
VBM4D	3.59	-
MS-EPLL	1618.90	-
3D-focus	127.33	-
MVCNN	29.33	-
LFBM5D	-	35.27
Proposed	-	1.80

**Table 8 sensors-21-02919-t008:** Comparison of processing time.

	Searching	Number of Patches
BM3D	100.60 × 10^7^	8.49 × 10^6^
VBM4D	951.58 × 10^7^	2.21 × 10^6^
MS-EPLL	-	-
3D-focus	2.06 × 10^7^	98.30 × 10^6^
MVCNN	-	-
LFBM5D	7.86 × 10^7^	10.78 × 10^6^
Proposed	0.06 × 10^7^	4.34 × 10^6^

## Data Availability

The codes used for the proposed scheme and for performance comparison are uploaded to https://github.com/Digital-System-Design-Lab/Compression-Aware-Multi-view-Denoising.git (accessed on 21 April 2021).

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
