# Peer review of "Improved Light Field Compression Efficiency through BM3D-Based Denoising Using Inter-View Correlation"

_sensors, 2021, doi:10.3390/s21092919_

Round 1

Reviewer 1 Report

This paper is well-structured and well-written, which is the good fit for the scope. 

Author Response

Thank you for your review efforts. I have sincerely addressed your concern in the attached file.

Reviewer 2 Report

This paper is well written but a minor revision is requested based on the following comments:

- in formula 1, why λmotion represents the Lagrangian multiplier?

- please  explain better this phrase at page 6 which appears unclear: “Unlike general block matching in many denoising algorithms, this 235 paper adopts RDVD in the cost calculation in order to reflect the spatial correlation by which 236 pixels belonging to the block have the same MVs in the block-based ME. Also, a point 237 with a small RDVD value is preferred to follow the condition in ME which considers the 238 rate-distortion trade-off as shown in (1).” Give a better version

- in formula 2, what is the coefficient ?

- supply the formula of PSNR and BDBR used in Table 1

- which formula is applied for the additive white   gaussian noise (AWGN) of sigma? such formula is similar in 3D case and 2D case?

Author Response

(The authors gave the same response as above.)

Reviewer 3 Report

This paper presents a denoising method utilizing structural information of the image along with the compression technique.  The paper is well written and contribute a significant improvement, but I would suggest some addition for the better understanding of the readers.

  1. Line 37, please highlight the reason of diverse pattern of noise in Multiview images and why it is more difficult to denoise them?
  2. I would suggest summarizing the state of the art in the form of table. Although sufficient background is given but the explanation is confusing. Summarizing them in table is highly recommended.
  3. Please specify neighboring views limit in line 162 and for search window in line 165.
  4. Line 167 define grouping criteria.
  5. How did you set the limit of threshold and what is the effect on the performance of the method?
  6. Methods in section 2 are not clearly explained. Several minor details are missing. I would suggest expanding this section and explain each step-in figure 1 in detail.

Author Response

(The authors gave the same response as above.)

Round 2

Reviewer 3 Report

Most of the highlighted comments are addressed in the revise version. I recommend to accept the paper.